# The Clinical Relevance of Target Lymph Node Biopsy after Primary Systemic Therapy in Initially Node-Positive Breast Cancer Patients

**DOI:** 10.3390/cancers13112620

**Published:** 2021-05-26

**Authors:** Steffi Hartmann, Angrit Stachs, Gesche Schultek, Bernd Gerber, Toralf Reimer

**Affiliations:** Department of Obstetrics and Gynecology, University of Rostock, 18059 Rostock, Germany; angrit.stachs@kliniksued-rostock.de (A.S.); gesche.schultek@kliniksued-rostock.de (G.S.); bernd.gerber@kliniksued-rostock.de (B.G.); toralf.reimer@kliniksued-rostock.de (T.R.)

**Keywords:** breast cancer, target lymph node, primary systemic therapy, sentinel lymph node, targeted axillary dissection

## Abstract

**Simple Summary:**

Currently, the optimal axillary surgical approach for breast cancer patients with initial node-positive disease and conversion to clinically node-negative status after primary systemic therapy is unclear. The aim of our study was to evaluate the clinical impact of removing the initially most suspicious, labeled axillary lymph node in addition to the sentinel lymph node. Metastatic target lymph nodes were found in five out of 63 patients (7.9%), while the sentinel lymph node was either tumor-free or not detected. The removal of the target lymph node influenced the adjuvant systemic therapy in only one case (1.6%). However, complete axillary dissection was indicated in all five cases. Furthermore, with fewer than three sentinel lymph nodes removed, the target lymph node reduced the false-negative rate to less than 10%. We therefore conclude that although the target lymph node has a minor impact on adjuvant systemic therapy, it is relevant for surgical axillary management.

**Abstract:**

Purpose: To assess the impact of the removal of the target lymph node (TLN) on therapy after the completion of primary systemic therapy (PST) in initially node-positive breast cancer patients. Methods: Pooled data analysis of participants of the prospective CLIP- and TATTOO-study at the University of Rostock was performed. Results: A total of 75 patients were included; 63 of them (84.0%) converted to clinically node-negative after PST. Both TLN and sentinel lymph node (SLN) were identified in 41 patients (51.2%). In five out of 63 patients (7.9%), the TLN was metastatic after PST and the SLN was either tumor-free or not detected. Axillary lymph node dissection (ALND) was conducted in all five patients. In one patient, systemic therapy recommendation was influenced by the TLN; adjuvant radiotherapy was influenced by the TLN in zero patients. For patients with fewer than three removed SLNs, the FNR was 28.6% for the SLN biopsy alone and 7.1% for targeted axillary dissection (TAD). Conclusions: Removal of the TLN in addition to the SLN after PST has only minimal impact on the type of adjuvant systemic therapy and radiotherapy. However, the extent of axillary surgery was relevantly affected and FNR was improved by TAD.

## 1. Introduction

Currently, the optimal axillary surgical approach for breast cancer patients with initial node-positive disease and conversion to clinically node-negative status after primary systemic therapy (PST) is unclear. The various international guidelines provide very different recommendations in this issue [1]. On the one hand, depending on tumor biology, 40–74% of these patients achieve a pathologically negative nodal status by PST [2,3,4], and therefore are candidates for less radical axillary surgery such as sentinel lymph node biopsy (SLNB), which is less likely to lead to complications such as pain, shoulder dysfunction, paraesthesia, or lymphedema compared with complete axillary lymph node dissection (ALND) [5,6]. On the other hand, an unacceptably high false-negative rate (FNR) of 17% has been determined for SLNB alone in this population [7]. This high FNR can be reduced to below 10%, which is generally considered acceptable, in different ways. In the multicenter prospective studies SENTINA [8] and ACOSOG Z1071 [9], FNR was less than 10% when at least three dual radionuclide and dye-labeled SLNs were removed. Furthermore, in the SN FNAC study, the FNR dropped from 13.3 to 8.4% if isolated residual tumor cells detected by immunohistochemistry in the lymph nodes were scored as node-positive [10]. In a retrospective subgroup analysis of 141 participants in the ACOSOG Z1071 study, in which the initially most suspicious axillary lymph node was marked with a clip, a reduction in the FNR of the SLN to 6.8% was shown when the clip-marked lymph node matched the removed SLN. In contrast, if the clip-labeled lymph node was not an SLN, the FNR was 19% [11]. Subsequently, several studies were performed to mark and selectively remove the initially most suspicious so-called target lymph node (TLN). Labeling the TLN before PST and removing it after PST (target lymph node biopsy (TLNB)) together with the SLN, is defined as targeted axillary dissection (TAD) [12]. The largest published prospective, multicenter study of TAD to date, the SenTa study, included 423 initially node-positive breast cancer patients and determined a low FNR for a TAD of 4.3% [13]. Because more than two SLNs can be detected in only 34 [8] to 56% [9] of cases, TAD is an option for patients with fewer than three detectable SLNs to forgo ALND in the setting of complete pathologic axillary remission. Recently, a study presented at the 2020 San Antonio Breast Cancer Symposium (SABCS) showed that, in 104 breast cancer patients with clinical nodal downstaging after PST, removal of the TLN in addition to SLNB did not result in a change in adjuvant systemic therapy recommendation in any case. However, at least three SLNs were removed in 83% of the cases analyzed and the concordance rate (CR) of TLN and SLN was 83% [14]. In the German CLIP-study, on average, only 1.6 SLNs were removed, resulting in a CR of 35.7% for SLN and TLN [15], and in the European multicentric TATTOO study a CR of only 59.7% was determined [16]. Therefore, this study aimed to evaluate the impact of removing the TLN in addition to SLN on adjuvant therapy and FNR in a patient population in which the TLN more often does not correspond to the SLN and a lower number of SLNs is removed.

## 2. Materials and Methods

### 2.1. Patient Population

Pooled data of patients included in the CLIP-study (DRKS00009793) and the TATTOO-study (DRKS00013169) at the breast care unit of the University of Rostock, Germany are presented. The local ethics committee approved these prospective feasibility trials, whose primary study objective was to determine the detection rate (DR) of the TLN after PST. Written informed consent was obtained from all participants. Patients presenting with histologically proven invasive breast cancer and clinically and/or sonographically suspicious ipsilateral axillary lymph nodes undergoing PST were included. Exclusion criteria were distant metastatic disease, pregnancy, and age ˂18 years.

### 2.2. Labeling of the TLN and Axillary Surgery

In both studies, the TLN was labeled before PST and targeted and removed after PST. In the CLIP-study, the TLN was marked with a clip (HydroMARK, Devicor Medical, Norderstedt, Germany, Reference # 4010-02-15-T3) under ultrasound-guidance and localized with a wire (Somatex Duo-System correctable localization kit, Somatex Medical Technologies GmbH, Teltow, Germany) immediately before operation with mammographic or sonographic support. In the TATTOO-study, the TLN was tattooed using a carbon solution (Spot^®^ GI Supply, Camp Hill, PA, USA, or CARBO-REP^®^ Sterylab, Rho/Milan, Italy) under ultrasound guidance, and detected and removed purely visually, intraoperatively. In patients still presenting with clinically or sonographically suspicious axillary lymph nodes after PST (ycN+), ALND, including levels I and II, was performed. In patients without suspicious lymph nodes after PST (ycN0), axillary SLNB by radiocolloid, blue dye, or dual mapping was conducted. In these cases, ALND was mandatory for patients undergoing surgery until March 2019 and optional after that timepoint according to the current guidelines of the German Gynecological Oncology Group (AGO) [17].

### 2.3. Pathologic Eevaluation

All lymph nodes were assessed by routine histopathological evaluation using hematoxylin and eosin staining, immunohistochemical staining for isolated tumor cells was not required by the study protocols. Pathologic complete response (pCR) after PST was defined as the absence of any invasive tumor in the breast and axillary lymph nodes [18].

### 2.4. Statistical Analysis

Descriptive statistics were computed for variables of interest and are presented as mean ± standard deviation (SD) and ranges for continuous data and frequencies and relative frequencies for categorical data, respectively. DR was the proportion of patients in whom the marked TLN or SLN was detected and removed during axillary surgery after PST. CR denotes the proportion of cases in which the TLN and SLN were matched. The proportion of patients with negative TAD nodes despite affected axillary lymph nodes of level I/II, out of all patients with affected lymph nodes (TLN, SLN, and/or ALND) after PST, was reported as FNR. The two-sample *t*-test for independent samples was used to test quantitative data for significant differences. The *p* values resulted from two-sided statistical tests and values of *p* ˂ 0.05 were regarded as statistically significant. SPSS/PC software package version 27 (IBM, Armonk, NY, USA) was used for data analysis.

## 3. Results

### 3.1. Demographic and Clinicopathologic Characteristics

Data from all patients enrolled in the CLIP-study (*n* = 30) and the TATTOO-study (*n* = 45) at the University of Rostock between February 2016 and October 2019 were analyzed, resulting in 75 datasets. Clinicopathological characteristics are presented in Table 1. Mean age was 55.1 ± 12.4 years (range 24–79 years) and ultrasound before PST revealed a mean tumor size of 27.8 ± 10.6 mm (range 8–57 mm). A chemotherapy regimen combining anthracyclines and taxanes was administered in 65 out of 75 patients (86.7%). Taxanes alone were given in 9.3% of patients (seven out of 75). PST was prematurely discontinued in three patients, in one patient after three cycles of anthracycline due to toxicity, and two patients after two cycles of anthracycline due to progressive disease. Anti-HER2-targeted therapy combining trastuzumab and pertuzumab was given to all 25 patients (100%) presenting with HER2-positive disease. In 50.0% of TNBC (17 out of 34), platinum was added to the chemotherapy regimen.

### 3.2. DR and FNR of SLN and TLN

The axillary TLN could be successfully removed after PST in 60 out of 75 patients (DR 80.0%). As depicted in Figure 1, in 63 out of 75 patients (84.0%), axillary lymph nodes became clinically and sonographically negative (ycN0) after PST. SLN mapping was performed in 60 of these 63 patients (95.2%). Both TLN and SLN were identified in 41 patients and were congruent in 21 of those (CR 51.2%). The mean number of TAD nodes was significantly higher (*p* = 0.001) in the group with discordant TLN and SLN (3.35 ± 1.46) compared to the group with concordant TAD nodes (1.9 ± 1.04). At least one SLN was detected in 50 patients (DR 83.3%). Eight out of these 50 patients (16.0%) with detectable SLN had at least three SLNs removed, whereas 42 out of 50 patients (84.0%) had fewer than three SLNs removed. The mean number of SLNs removed was 1.72 ± 0.95 (range 1–5). ALND was performed in 65 out of 75 patients (86.7%) with a mean number of removed lymph nodes of 10.92 ± 3.83 (range 4–23). In the group of patients without ALND (10 out of 75 patients, 13.3%), the mean number of removed lymph nodes was significantly lower (*p* = 0.001) with 2.20 ± 1.03 (range 1–4).

As shown in Table 2, any residual lymph node metastases were detected in 20 ycN0 patients in which ALND was performed. The FNR in this subgroup was 5.0% for TAD (one out of 20), 15.0% for TLNB (three out of 20), and 20.0% for SLNB (four out of 20). For the 14 out of 42 patients (33.3%) with metastatic lymph nodes after PST, successful SLNB and fewer than three removed SLNs, the FNRs were 28.6% for SLNB, 14.3% for TLNB, and 7.1% for TAD (Table 2). No false-negative results for SLNB, TLNB, or TAD were found if at least three SLNs were removed. Five out of 20 patients (25.0%) with a TLN that was not congruent with the SLN had metastatic lymph nodes after PST and, in three cases, SLN and TLN were negative (FNR 60.0%), the FNR of TAD was 20.0% (one out of five) (Table 2). Eight out of 21 patients (38.1%) with concordant TLN and SLN had positive lymph nodes after PST, none of them with a false-negative SLNB, TLNB, or TAD (FNR 0.0%).

### 3.3. Impact of TLN on Treatment

In five out of 63 ycN0 patients (7.9%), the TLN was metastatic after PST and the SLN was either tumor-free or not detected (Table 3). ALND was performed in all five cases and adjuvant radiotherapy, including lymphatic drainage area, was recommended. Both cases with negative SLN were patients with hormone receptor-positive, HER2-negative breast carcinoma. Anti-hormonal therapy was recommended for them in the post-surgery tumor board. The three patients with affected TLN and undetected SLN had tumors with different tumor biology. One patient with TNBC showed no residual tumor in the breast but three metastatic axillary lymph nodes and, therefore, no pCR. Thus, postneoadjuvant chemotherapy with capecitabine was recommended in the tumor board. Another patient with a hormone receptor-negative, HER2-positive tumor had a 5 mm invasive residual tumor in the breast and two metastatic lymph nodes. The woman continued to postoperatively receive a dual HER2 blockade with trastuzumab and pertuzumab. The fifth patient with hormone receptor-positive, HER2-negative breast carcinoma, 11 mm invasive residual tumor in the breast, and five metastatic lymph nodes was recommended for anti-hormonal therapy. In summary, a switch in postoperative systemic therapy due to the removal of the TLN in addition to the SLN was observed in one out of 63 cases (1.6%).

## 4. Discussion

In the current study, additional removal of the TLN to the SLN after PST in breast cancer patients with initially suspicious axillary lymph nodes had an impact on the postoperative recommendation regarding systemic therapy in only one of 63 cases. This confirms the results of a prospective study presented at SABCS 2020, in which TLNB in addition to SLNB did not result in a change in adjuvant systemic therapy in 104 patients [14]. Contrary to our assumption, the comparatively high proportion of discordant TLN and SLN of 48.8% in the current study and the significantly higher number of removed TAD nodes in these cases also did not result in a greater impact of TLN on systemic therapy after surgery. The small impact of TLNB on adjuvant systemic therapy may be explained by the fact that the current recommendation is dependent on the achievement of a pCR, and on tumor biology. Patients with HER2-positive breast cancer who still have metastatic axillary lymph nodes after PST benefit significantly from an adjuvant switch from anti-HER2 therapy to trastuzumab emtansine (T-DM1) in terms of invasive disease-free survival (DFS) compared with continuation of trastuzumab therapy started neoadjuvantly [19]. Furthermore, patients with HER2-negative breast cancer benefit from postneoadjuvant capecitabine if they do not present with pCR, especially the subgroups with TNBC and one to three metastatic lymph nodes after PST [20]. However, we know that especially in patients with TNBC and HER2-positive breast cancer, PST achieves pCR in up to 74%, which includes tumor-free axillary lymph nodes [2,3]. Furthermore, in patients with this tumor biology, even with initially node-positive breast cancer, if no tumor cells are detectable in the breast after PST, lymph node metastases can no longer be detected in 89.6% of cases [21]. Although the overall response to PST in invasive lobular breast carcinoma is poorer than in invasive ductal tumor type, a better response in TNBC and HER2-positive tumors, and a survival benefit when a pCR is achieved, can also be demonstrated in invasive lobular carcinoma [22]. In contrast, according to the current German AGO recommendation, the presence of a pCR, and thus also the axillary lymph node status in patients with hormone receptor-positive, HER2-negative breast cancer, does not influence the type of adjuvant systemic therapy [17]. Therefore, the overall proportion of patients with axillary lymph nodes still affected after PST without residual tumor in the breast and with TNBC or HER2-positive subtype is assessed as very low, thus explaining the minor influence of TLN on systemic therapy.

However, irrespective of the influence on systemic therapy, the affected TLN in the presence of negative or undetected SLN had an influence on surgical therapy in the axilla in all five cases. Indeed, ALND is currently recommended for these patients in addition to TAD [17]. The background for this recommendation is that 64% of patients with micrometastases and 62% of patients with macrometastases in the SLN have additional lymph node metastases after PST [23]. The role of regional radiotherapy in initially node-positive patients is currently being investigated in two ongoing randomized phases III trials, the Alliance A11202 trial (NCT01901094) [24] and the NSABP B-51 trial (NCT01872975) [25]. The Alliance A11202 study is investigating whether radiotherapy of the axilla is a safe alternative to ALND in initially node-positive breast cancer patients with affected SLN after PST. Results are expected in 2024 [24]. In contrast, the NSABP B-51 trial is evaluating whether regional nodal radiation therapy is necessary for patients who no longer have affected axillary lymph nodes after PST. Evaluation of the primary study objective is planned for 2023 [25]. Until these results are available, the indication for postmastectomy radiotherapy and regional nodal irradiation is based only on the initial tumor stage and tumor biology, and in all five patients, radiotherapy would have been recommended in the same way after PST, even without the involvement of the TLN, according to the German guidelines [17].

No false-negative results for SLN were detected in the group of patients with at least three removed SLNs, supporting the NCCN recommendation for SLNB alone in cases of at least three excised SLNs [26]. However, the proportion of patients with more than two SLNs in this study is comparatively low, at only 16%, and it was possible to reduce the FNR from 28.6% to less than 10% in the group of patients with less than three SLNs removed by additional removal of the TLN. It can be concluded that in patients with less than three SLNs, the additional removal of the TLN keeps the FNR within an acceptable range, and thus spares them the need for ALND. The reason for the low proportion of patients with more than two removed SLNs in the current study is probably the definition of SLN. While in the ACOSOG-Z1071-study [9], as well as in the American prospective pilot study on TAD [12], all marked and palpably suspicious lymph nodes were declared as SLNs, in the current study only the marked lymph nodes were removed as SLNs, and palpably suspicious and unmarked lymph nodes were assigned to level I or II. However, the generally accepted FNR of a maximum of 10% for SLNB or TAD is a hypothetical limit and currently no data are available regarding how a higher or lower FNR affects the prognosis of patients. To date, only retrospective data on prognostic relevance after PST of persistent axillary lymph node metastases have been published. Wong et al. reported a significantly worse 5-year overall survival (OAS) for patients with lymph node metastases (60.6–81.0%) compared with patients without lymph node metastases (86.7%) after PST in more than 35,000 patients from the American National Cancer Database [27]. Furthermore, in a multicenter study for patients with initial node-positive disease and exclusive SLNB with clinical nodal downstaging after PST (less than three SLNs removed in 74.3% of cases), a low axillary recurrence rate of 1.7% was observed after ten years of follow-up and no worsening of distant DFS and OAS were detected compared to initial clinical node-negative patients [28]. Another retrospective study showed significantly better DFS and OAS in 59 patients with clinical nodal downstaging after PST for study participants with SLNB alone in tumor-free SLN, compared with patients who additionally received ALND in metastatic SLN [29]. Prospective data on the impact of the type of axillary surgery on the prognosis of initial node-positive breast cancer patients treated with PST are urgently needed. Two prospective, multicenter registry studies are currently investigating whether FNR and the type of axillary surgery have an impact on disease progression in breast cancer patients with initially suspicious axillary lymph nodes after PST. Results from the Dutch MINIMAX study (NCT04486495) are expected in 2027 [30]. The 5-year invasive DFS data and axillary recurrence rate from the European AXSANA study (NCT04373655), depending on the type of axillary surgery, will be available in 2030 [1].

## 5. Conclusions

Even in a study cohort with low CR for TLN and SLN and a low proportion of patients with more than two SLNs removed, removal of the TLN in addition to the SLN in patients with the clinical conversion of axillary lymph node status after PST has minimal impact on the type of adjuvant systemic therapy and radiotherapy. Due to the small number of included patients, these results should be further clarified in larger, multicentric trials. However, the additional TLNB lowered the FNR to an acceptable range in patients with less than three SLNs removed, and it resulted in the recommendation for ALND in all cases with undetected or non-metastatic SLN and tumor cell detection in the TLN after PST. Whether radiotherapy of the axilla is an alternative to ALND in these cases and what role FNR plays in disease progression will be shown by the results of future prospective, multicenter studies.

## Figures and Tables

**Figure 1 cancers-13-02620-f001:**
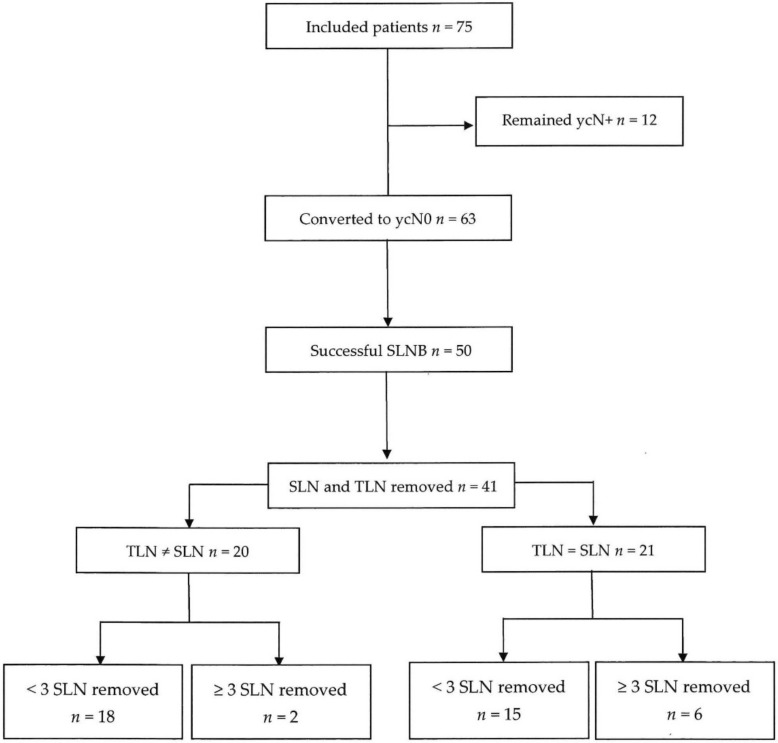
Flow chart for the trial. Abbreviations: SLNB—sentinel lymph node biopsy, TLN—target lymph node, SLN—sentinel lymph node.

**Table 1 cancers-13-02620-t001:** Clinicopathologic characteristics of the study cohort (*n* = 75).

Characteristics	No. of Patients	%
Initial clinical tumor stage		
cT1	19	25.3
cT2	53	70.7
cT3/4	3	4.0
No. of initially suspicious LN on US		
1–3	62	82.7
˃3	13	17.3
Tumor type		
invasive ductal	66	88.0
invasive lobular	5	6.7
other	4	5.3
Grading		
2	16	21.3
3	59	78.7
Receptor-based subtype		
HR+/HER2−	16	21.3
HR+/HER2+	11	14.7
HR−/HER2+	14	18.7
HR−/HER2−	34	45.3
Suspicious LN on US after PST		
no	63	84.0
yes	12	16.0
Pathologic complete remission		
no	44	58.7
yes	31	41.3
Type of surgery		
breast conservation	56	74.7
mastectomy	19	25.3

Abbreviations: LN—lymph node, US—ultrasound, HR—hormone receptor, HER2—human epidermal growth factor receptor 2, PST—primary systemic therapy.

**Table 2 cancers-13-02620-t002:** False-negative results of TLNB, SLNB, and TAD.

Removed LN	ycN0/*n* = 20 (%)	No. of SLN ˂3/*n* = 14 (%)	SLN ≠ TLN/*n* = 5 (%)
SLNB	4 (20.0)	4 (28.6)	3 (60.0)
TLNB	3 (15.0)	2 (14.3)	3 (60.0)
TAD	1 (5.0)	1 (7.1)	1 (20.0)

Abbreviations: LN—lymph node, TLNB—target lymph node biopsy, SLNB—sentinel lymph node biopsy, TAD—targeted axillary dissection, SLN—sentinel lymph node, TLN—target lymph node.

**Table 3 cancers-13-02620-t003:** Clinicopathological characteristics of patients with metastatic TLN and tumor-free or not detected SLN after PST (*n* = 5).

Patient	1	2	3	4	5
Age (years)	53	45	66	73	63
Initial tumor size (mm)	47	23	57	17	17
No. of initially suspicious LN	1	2	3	2	1
Receptor-based subtype	HR+/ HER2−	HR+/ HER2−	HR−/HER2+	HR−/HER2−	HR+/ HER2−
Type of breast surgery	ME	BCS	ME	BCS	BCS
Invasive breast tumor after PST (mm)	8	12	5	0	11
metastatic TLN	1/1	1/2	1/1	1/2	2/3
SLN	0/1	0/1	ND	ND	ND
metastatic LN (TLN + ALND)	2/13	1/8	2/6	3/7	5/8

Abbreviations: TLN—target lymph node, SLN—sentinel lymph node, PST—primary systemic therapy, LN—lymph node, HR—hormone receptor, HER2—human epidermal growth factor receptor 2, PST— primary systemic therapy, ME—mastectomy, BCS—breast conserving surgery, ND—not detected, ALND—axillary lymph node dissection.

## Data Availability

The data presented in this study are available on request from the corresponding author.

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
