# Peer review of "The Clinical Relevance of Target Lymph Node Biopsy after Primary Systemic Therapy in Initially Node-Positive Breast Cancer Patients"

_cancers, 2021, doi:10.3390/cancers13112620_

Round 1

Reviewer 1 Report

This manuscript reported the clinical importance of TLNB after NAC in clinical node positive BC patients. I think that this clinical question is important and I understood the authors' opinion. However the number of patients was too small to confirm their hypothesis. Moreover, there were two major points which influence the results. One is diagnosis of clinical lymph node metastasis, especially the number of metastatic nodes. Another is the number of detected SLN. In this report, there was no details about of them. We know that It is very difficult to decide them. So we need to enough number of patients to discuss about this clinical question. To confirm the residual invasive disease after NAC become more and more important for appropriate decision making about adjuvant therapy.

Author Response

Thank you very much for these important comments. The authors have now included a sentence into the conclusion section regarding the small number of included patients.

The authors are aware of the importance of initially affected lymph nodes and the number of detected SLN for the interpretation of the studies results. Information on the number of initially sonographically suspicious lymph nodes can be found in Table 1 and has therefore not been repeated in the text. Data on the number of SLNs detected/removed has now been added to the results section.

Reviewer 2 Report

The ms titled "The clinical relevance of target lymph node biopsy after primary systemic therapy in initially node-positive breast cancer patients" describes the use of lymph node biopsy in breast cancer clinical relevance.

The ms is interesting, with good results and methods well described. The data open the way to a new approach in clinic.

Nevertheless, I think that the introduction is of poor quality, not highlighting the impact of results later proposed. Please add other information in order to drive the reader into the results.

A question also is : why the authors have not included other markers such as some Gpcrs or kinases in tnbc classification. This could be beneficial.

Add a figure as a sort of graphical summary, to rationalize the data. 

Author Response

Comment: Nevertheless, I think that the introduction is of poor quality, not highlighting the impact of results later proposed. Please add other information in order to drive the reader into the results.

Response: Thank you very much for this important comment. We have extensively revised the introduction to emphasize the clinical significance of targeted axillary dissection.

Comment: A question also is : why the authors have not included other markers such as some Gpcrs or kinases in tnbc classification. This could be beneficial.

Response: The authors did not include these markers in the analysis because 55% of the study participants did not have triple-negative breast carcinoma and therefore an additional analysis in the triple-negative subgroup would not have yielded valid results and, secondly, there are not yet even prospective data on the influence of TAD on the prognosis of all initial nodal-positive patients and therefore not on the influence of GPCRs and kinases in the TNBC subpopulation. However, this should be the subject of larger, prospective, multicenter studies in the future. Outcomes of the submitted analysis were not insights into breast tumorigenesis like GPCR- or kinase based signals.  

Comment: Add a figure as a sort of graphical summary, to rationalize the data.

Response: A graphical abstract was added.

Reviewer 3 Report

This study adresses a very important issue concerning surgery for brast cancer and the possibility to de-escalate surgery in the axilla. As the manuscript points out ALND is associated with pain and morbidity and reduced quality of life. Hence if we can de-escalate the surgery in the axilla with the similar good results as we have today, this will benefit the patient and the health expenses in the society. The study describes how this safely can be done with the lowest possible false negative rate. Important prospective studied are mentioned and results from these studies are awaited with great interest.

There are a few issues to be further elucidated.

  1. The difference between TLND and TAD is not perfectly clear to the reader and especially one that is not familiar with this kind of surgery. It needs to be specified.
  2. Table 2 needs to be explained better. It is not obvious what the author is trying to illustrate here
  3. Lobular carcinoma reacts to neoadjuvant chemotherapy in an unpredictable manner with variable response and appearance. This issue should be discussed

Author Response

 There are a few issues to be further elucidated.

1. The difference between TLND and TAD is not perfectly clear to the reader and especially one that is not familiar with this kind of surgery. It needs to be specified.

Response: In the introduction, TLNB was defined and the difference to TAD was made clearer.

2. Table 2 needs to be explained better. It is not obvious what the author is trying to illustrate here

Response: The paragraph in the results section explaining the results from Table 2 has been rewritten to make the information in the table easier to understand.

3. Lobular carcinoma reacts to neoadjuvant chemotherapy in an unpredictable manner with variable response and appearance. This issue should be discussed

Response: A section on invasive lobular breast carcinoma and response to PST was added to the discussion.

Round 2

Reviewer 1 Report

I think that this manuscript include some biases and problems to confirm their hypothesis. However I think that these results were valuable for next research.